**DOI: 10.1038/ncomms13139**　　**OPEN**

# Self-assembly of acetate adsorbates drives atomic rearrangement on the Au(110) surface

Fanny Hiebel[1], Bonggeun Shong[1,†], Wei Chen[2,3,4], Robert J. Madix[2], Efthimios Kaxiras[2,3] & Cynthia M. Friend[1,2]

Weak inter-adsorbate interactions are shown to play a crucial role in determining surface structure, with major implications for its catalytic reactivity. This is exemplified here in the case of acetate bound to Au(110), where the small extra energy of the van der Waals interactions among the surface-bound groups drives massive restructuring of the underlying Au. Acetate is a key intermediate in electro-oxidation of $CO_2$ and a poison in partial oxidation reactions. Metal atom migration originates at surface defects and is likely facilitated by weakened Au–Au interactions due to bonding with the acetate. Even though the acetate is a relatively small molecule, weak intermolecular interaction provides the energy required for molecular self-assembly and reorganization of the metal surface.

[1] Department of Chemistry and Chemical Biology, Harvard University, Cambridge, Massachusetts 02138, USA. [2] School of Engineering and Applied Science, Harvard University, Cambridge, Massachusetts 02138, USA. [3] Department of Physics, Harvard University, Cambridge, Massachusetts 02138, USA. [4] International Center for Quantum Design of Functional Materials (ICQD), Hefei National Laboratory for Physical Sciences at Microscale, and Synergetic Innovation Center of Quantum Information and Quantum Physics, University of Science and Technology of China, Hefei, Anhui 230026, China. † Present address: Department of Chemistry, Chungnam National University, Daejeon 34134, South Korea. Correspondence and requests for materials should be addressed to C.M.F. (email: friend@fas.harvard.edu).

A critical aspect of designing efficient heterogeneous catalytic reactions is the determination of factors responsible for the relative stability of intermediates on the surface of the catalyst. This relative stability dictates the relative concentrations of various species on the surface and the selectivity of a given reaction. The presence of highly stable species, which do not lead to productive reactions, inhibits desired reactions and leads to the suppression of catalytic activity. The stability of an intermediate is related to the strength of its binding to the catalyst surface; the contribution of weak interactions to this stability can play a decisive role and even modify the relative stability of the species[1].

The Au surface is a model for understanding oxidative processes important in heterogeneous catalysis. Gold-based catalysts are of fundamental interest because of their potential for highly selective oxidation reactions[2–4], including selective oxidation of alcohols[5,6]. Single-crystal model studies have successfully established frameworks for understanding the catalytic behaviour of Au (refs 7,8) by comparison to functional catalysts and in particular nanoporous gold[9,10]. Acetate has specifically been identified as an intermediate in ethanol oxidation that leads to combustion[11]. Acetate is very strongly bound to both Au(111) (ref. 1) and Au(110) (Supplementary Table 1 and Supplementary Discussion) and decreases overall chemical activity by inhibiting the formation of key intermediates, for example, ethoxy, for selective oxidation. It is an example of an important class of molecular species, carboxylates, that bind strongly to surfaces and that are key intermediates in the cycle of $CO_2$ production and remediation (Fig. 1). Carboxylates lead to undesirable combustion to $CO_2$ in catalytic oxidation reactions[11–15], and at the same time are possible intermediates in the reverse process of $CO_2$ reduction[16,17].

Here we report on a system where the contribution of weak interactions between adsorbates affects the stability of the adsorbed intermediate, acetate, so as to induce structural modifications of the catalyst surface. We demonstrate through a combination of molecular-scale imaging, reactivity studies and density functional theory (DFT) calculations that acetate condenses into structures with high local concentration (~0.25 monolayer (ML)) even when the global concentration is low (~0.05 ML) on Au(110) at room temperature. Significant surface roughening indicates that Au atoms from the clean surface are harnessed to enable the process of acetate self-assembly, a process facilitated by surface defects. The thermodynamic stability of the densely packed acetate phase is confirmed by the formation of large, highly ordered areas on mild heating. The DFT calculations reveal that weak inter-adsorbate interactions play a major role in the self-assembly of the acetate molecules and the required Au surface restructuring.

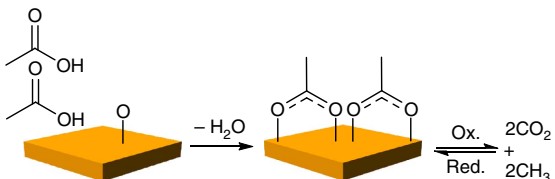

**Figure 1 | The adsorption of acetate by oxidation of acetic acid on an O-precovered surface and reduction of $CO_2$.** Carboxylates bind strongly on surfaces and are intermediates in $CO_2$ production and remediation. Here acetate is adsorbed on Au after reaction of acetic acid with adsorbed atomic O.

## Results

**Acetate adsorption procedure.** Experimentally, the acetate was adsorbed at room temperature (RT) by acid-base reaction of excess acetic acid with pre-adsorbed atomic oxygen (Fig. 1). Pre-adsorbed atomic oxygen is essential in this process because the clean Au(110) surface cannot cleave the O–H bond in the acetic acid, which consequently desorbs molecularly above ~230 K (Supplementary Fig. 1 and Supplementary Discussion). In the conditions considered here, the hydroxyls formed upon deprotonation of the acid readily recombine to form gaseous water, leaving only the carboxylate on the surface[18]. The final acetate coverage is controlled by varying the initial O coverage; the reaction stoichiometry is such that two acetates form per each pre-adsorbed O atom.

**Acetate thermal stability.** Like on the Au(111) surface, acetate is highly stable on the Au(110) surface. The decomposition mechanisms of acetate have been previously studied in detail on Au(111) (refs 13,19) and we find the same mechanisms on Au(110). Acetate decomposes to $CO_2$ and other products above 500 K (Supplementary Fig. 2a and Supplementary Discussion), the desorption of products peaking at ~50 K higher than on Au(111). In the following, we concentrate on scanning tunnelling microscopy (STM) results acquired at RT, that is, well below the decomposition temperature.

**The condensed acetate phase.** STM reveals the formation of dense structures of acetate on Au(110) for all coverages investigated (Fig. 2a–c). Clean Au(110) exhibits a (1 × 2) surface reconstruction (Au-(1 × 2)), also called 'missing-row'[20,21], which is unaffected by O adsorption at coverages considered here[22]. It consists of alternate troughs and ridges in the [1–10] direction, which appear as bright lines in STM. The row contrast is vanishing progressively as the acetate coverage is increased. The acetate saturation coverage is obtained after reaction of ~0.12 ML initial O with excess acetic acid, showing a continuous layer of acetate on a roughened surface (Fig. 2c). Consistently, the acetate saturation coverage is 0.25 ML. The coverage is defined on the basis of the number of exposed Au atoms within one unit cell of the clean Au-(1 × 2) reconstruction. Hence, a coverage of four species per unit cell is defined as 1 ML. The acetate saturation coverage (0.25 ML) corresponds to one molecule per Au-(1 × 2) unit cell.

The densely packed acetate domains show a c(2 × 2) molecular arrangement, for all global coverages investigated (Fig. 2d,e). The local coverage that corresponds to this molecular arrangement is 0.25 ML. It is noteworthy that this dense ordering is observed even at global coverage as low as 0.05 ML, which is derived from the ratio of clean Au area (row contrast) and acetate-covered areas (bright domains) (Fig. 2a). The details of the c(2 × 2) are apparent in high-resolution images (Fig. 2d,e). In the [1–10] direction, every other acetate row is aligned with the pristine Au rows (see magnified area, Fig. 2d). The distance between protrusions in the row direction is $a' = 5.5 \pm 0.3$ Å, which is ~2 times the Au–Au distance of 2.88 Å in the Au(110)-(1 × 1) (Au-(1 × 1)) unit cell. Every other row is shifted by $a'/2$, forming a c(2 × 2) structure with respect to the ideal Au(110) surface. At saturation coverage, the same c(2 × 2) structure is observed everywhere on the surface (Fig. 2e), consistently with ~0.25 ML coverage deduced from the initial O coverage. The c(2 × 2) acetate structure has been reported on Cu(110) (refs 23–26) and Ni(110) (refs 27,28) and the molecule adsorption geometry is bidentate top, with the two oxygen atoms in acetate binding to two topmost Au atoms along the close packed rows. Our DFT calculations show the same preferred adsorption geometry on

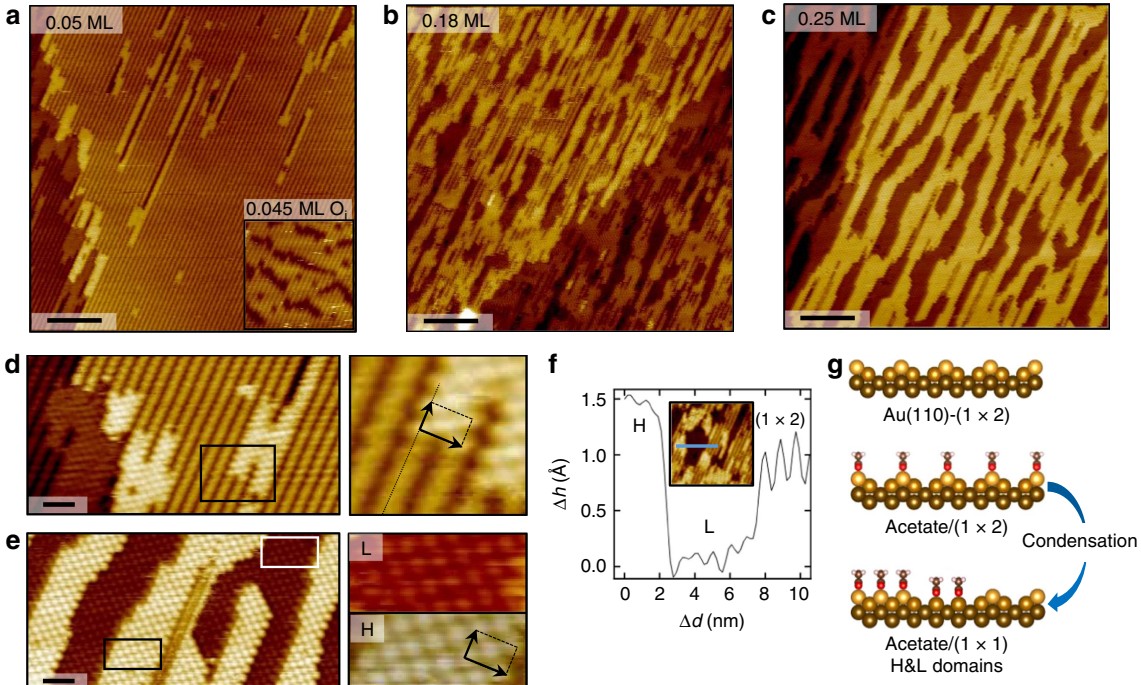

**Figure 2 | Acetate condensation into densely packed islands with a c(2 × 2) structure and concomitant Au interface bimodal roughening.** (**a–c**) Series of STM images of increasing acetate coverage up to saturation at 0.25 ML; Scale bar, 10 nm, sample bias: +0.5 to +1.5 V, tunnelling current: 0.1 to 1.0 nA. Inset in **a**: initial O-covered surface, same scale. The dark features are O chains[22]. (**d,e**) High resolution STM images reveal the c(2 × 2) molecular ordering of acetate over the entire coverage regime. Scale bar, 2 nm. Right panel: zoom-ins of the respective boxed areas. Arrows indicate the c(2 × 2) unit cell parameters. (**f**) Profile across the blue line in the inset (20 × 20 nm² image of the surface in **b**) spanning a high (H) and low (L) acetate-covered domain and the clean Au-(1 × 2) surface. (**g**) Side view of the Au-(1 × 2) to Au-(1 × 1) transformation necessary for acetate condensation explaining the surface bimodal roughening of the terraces detected by STM.

Au(110) (Supplementary Fig. 3 and Supplementary Table 2 and Supplementary Discussion). In the STM images, the bright protrusions correspond to the methyl group of the molecule, as shown by DFT simulations (Supplementary Fig. 4 and Supplementary Discussion).

**Adsorption-induced Au restructuring.** The terraces and step edges roughen upon acetate adsorption, as can be seen from the STM image series at increasing acetate coverage in Fig. 2a–c. This indicates the mobilization of Au atoms in the acetate condensation process. At saturation coverage, the terraces form a bimodal mosaic of high (H) and low (L) areas (Fig. 2c,e). At intermediate acetate coverage, H and L domains coexist with domains of pristine Au-(1 × 2) surface within the terraces. H domains are 0.5 ± 0.1 Å higher than clean Au-(1 × 2) areas and H and L areas differ in height by 1.4 ± 0.1 Å, which matches the Au(110) atomic step height (Fig. 2f).

The bimodal roughening of the surface can be interpreted as the transition from missing-row to ideal Au interface structure under the acetate layer (Fig. 2g). Acetate adsorbs on the ridges of the missing-row reconstruction. Upon condensation, migration of topmost Au atoms occurs to fill the missing-row of a neighbouring domain, resulting in alternate H and L domains of the same structure, composed of acetate on Au-(1 × 1).

**The role of surface defects in the acetate condensation.** The morphology of the condensed c(2 × 2) islands at low coverage (Figs 2 and 3) supports the Au-(1 × 2) to Au-(1 × 1) transition under the acetate layer and suggests that the nucleation is kinetically constrained to surface defects. We propose three elementary mechanisms for the acetate-induced Au restructuring,

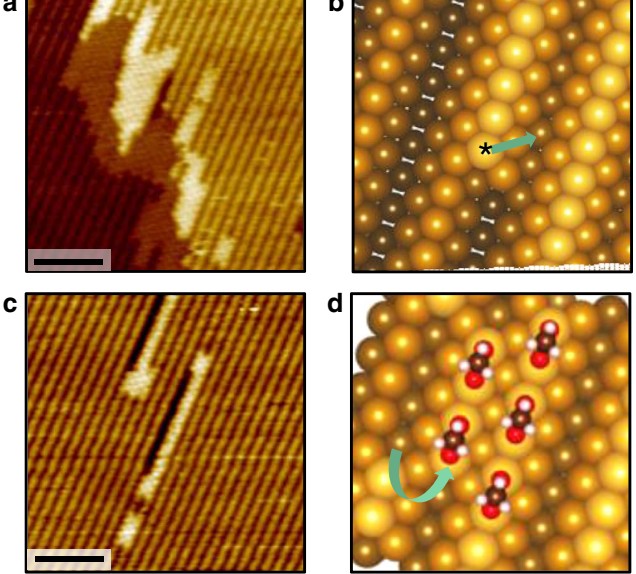

**Figure 3 | Acetate c(2 × 2) nucleation features and associated Au displacements.** (**a,c**) STM images of the surface in Fig. 2a (0.05 ML acetate) showing acetate islands around steps and linear features within the terrace. Scale bar, 5 nm. (**b**) Schematic of the undercoordinated atom at a step edge kink likely to migrate towards the upper terrace forming a local Au-(1 × 1) domain on both sides of the step, also accounting for mobile Au adatoms within terraces. (**d**) Schematic for row-pairing features within the terrace. Imaging parameters: Sample bias: +0.5 to +1.5 V, tunnelling current: 0.1 to 1.0 nA.

involving an acetate-Au complex. First, two-dimensional islands around steps (Fig. 3a) are attributed to step retraction starting from undercoordinated Au atoms (Fig. 3b). Second, one-dimensional structures, 10–20 nm long in the closest-packed direction within the terraces (Fig. 3c) are explained by Au row-pairing (Fig. 3d). Finally, isolated acetate islands are attributed to the capture of mobile Au adatoms present on the Au(110) surface during reaction.

The step retraction and row-pairing mechanisms are similar to the deconstruction mechanism of the metastable missing-row structure on Rh(110) (ref. 29), including the important role of defects such as undercoordinated metal atoms at kinks within step edges in the structural transition[30]. It is however important to note that in the cited study, the surface is adsorbate-free and the thermodynamics of the metal surface itself is the driving force for the reconstruction. Additionally here, the formation of molecule–metal complexes may further reduce the activation energy for extraction of undercoordinated atoms, as shown for CO on Au(110) (ref. 31). Regarding the Au adatom capture mechanism, it is known that undercoordinated Au atoms at kinks within step edges (Fig. 3b) are prone to detach and migrate onto the terraces, maintaining an equilibrium concentration of Au adatoms, even at RT (refs 32–34). The isolated acetate islands formed around trapped Au adatoms are in fact likely to constitute nucleation centres for the linear structure observed within the terraces.

**Metastable versus thermodynamic equilibrium surface structure.** The roughness of the surface at intermediate and saturation coverage can be related to the elementary deconstruction mechanisms identified at low coverage. The anisotropy of the pores observed at saturation coverage in Fig. 2c likely derives from the paired-rows structures identified at low coverage. Those structures introduce undercoordinated atoms at their terminations, which can constitute nucleation centres for two-dimensional acetate domains within the terraces. Interestingly, high mobility of acetate must occur right after formation, since the final distribution of acetate on the surface is much different from the distribution of atomic O (ref. 22) (see the inset in Fig. 2a), which are the active sites for acetic acid deprotonation. However, the observed final surface patterning can be explained by rather small displacements of acetate–Au complexes around surface defects. This suggests that the diffusion of the acetate-Au complexes is kinetically limited at RT. In the following, we investigate the thermodynamics of the system through mild annealing experiments.

When sufficient thermal energy is transferred to the system, large acetate domains are formed and the overall step density is strongly reduced (Fig. 4a). In this experiment, ∼0.15 ML acetate on Au was annealed below the acetate decomposition temperature (∼400 K) for 20 min. The high density of steps produced during RT acetate adsorption is strongly reduced, leaving rectangular pores of several tens of nm long on the terraces (arrow). Despite the observed large-scale reorganization, the c(2 × 2) molecular arrangement of acetate is preserved and extends over large areas (Fig. 4b). The extent of the acetate domains however appears limited to ∼10 nm in the direction of the Au rows; such limitation could arise from adsorption-induced accumulation of surface strain in the close-packed direction of the substrate[35]. Our DFT calculations reveal only small lateral distortions of the Au surface upon adsorption. The rather wide domains observed are consistent with only small strain effects. The above experimental observations show that as the temperature and thus the rate of diffusion increases, the density of steps decreases, indicating that the domain boundaries observed at RT are in fact metastable and they represent some

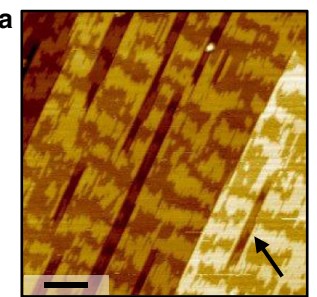
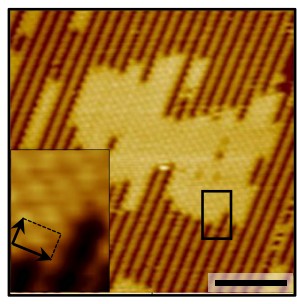

**Figure 4 | Equilibrium surface structure obtained upon mild annealing.** Heating ∼0.15 ML acetate for 20 min at ∼400 K induces reorganization of the acetate layer to a quasi-ordered striped structure and strongly reduces the atomic step density while maintaining the c(2 × 2) molecular ordering of acetate. STM images (**a**) Scale bar, 20 nm and (**b**) Scale bar, 5 nm, sample bias: +0.5 to +1.5 V, tunnelling current: 0.1 to 1.0 nA. The arrow in **a** shows a large pore resulting from step coalescence. The inset in **b** shows the c(2 × 2) acetate structure.

energy cost to the system. Closer to the thermodynamic equilibrium, large flat acetate domains are formed, even for a partially covered surface. Hence, the annealing experiment definitely eliminates the possibility of stronger molecule–substrate interaction with undercoordinated metal atoms as the driving force for reconstruction. In the following, DFT calculations are used to elucidate the molecular origin of the thermodynamic trend for acetate condensation.

**Molecular origin of the driving force for condensation.** We show here that the trend towards acetate condensation and interface deconstruction can only be reproduced by DFT calculations that take van der Waals (vdW) interactions into account. In this study, we consider three acetate molecular orderings of increasing local density ($\theta_{local} = 1/16$ ML, $\theta_{local} = 1/8$ ML and $\theta_{local} = 1/4$ ML), on Au-(1 × 2) and Au-(1 × 1). The stability of the structures was evaluated by defining an interface energy per acetate (equation (2)), which contains both the Au surface energy in the considered reconstruction and the acetate adsorption energy in the considered ordering. A constant global acetate concentration is constructed by considering linear combinations of the energetics of 4 × 2 supercells (Supplementary Fig. 5); as an example, we compare the sum of the interface energy of four supercells with $\theta_{local} = 1/16$ ML (Fig. 5a) to the sum of interface energy of one supercell with $\theta_{local} = 1/4$ ML (Fig. 5c) and three supercells of clean Au-(1 × 2) (Supplementary Fig. 5). With this descriptor, acetate layers of various local densities on various interfaces can be directly compared. Note that the cost of Au-(1 × 2) to Au-(1 × 1) transformation, calculated here to be 0.08 eV per 1 × 2 unit cell (3.3 meV Å$^{-2}$) with similar values found by previous studies[36,37], is already accounted for in this quantity.

For the two lower local coverages, the most stable structure is obtained on the Au-(1 × 2) interface (Fig. 5a,b) as the diluted phases cannot induce the Au interface transformation. On the other hand, the highest coverage, that is, the condensed phase, can only be achieved on the Au-(1 × 1) interface (Fig. 5c). This phase presents a lower interface energy per acetate than the diluted phases only when vdW contributions are considered (arrows in Fig. 5d). In terms of supercell energy, that is, four acetate molecules, this corresponds to +0.45 eV uphill with PBE only and −0.08 eV downhill with the corrected PBE + vdW, revealing their importance in describing the thermodynamics of the system.

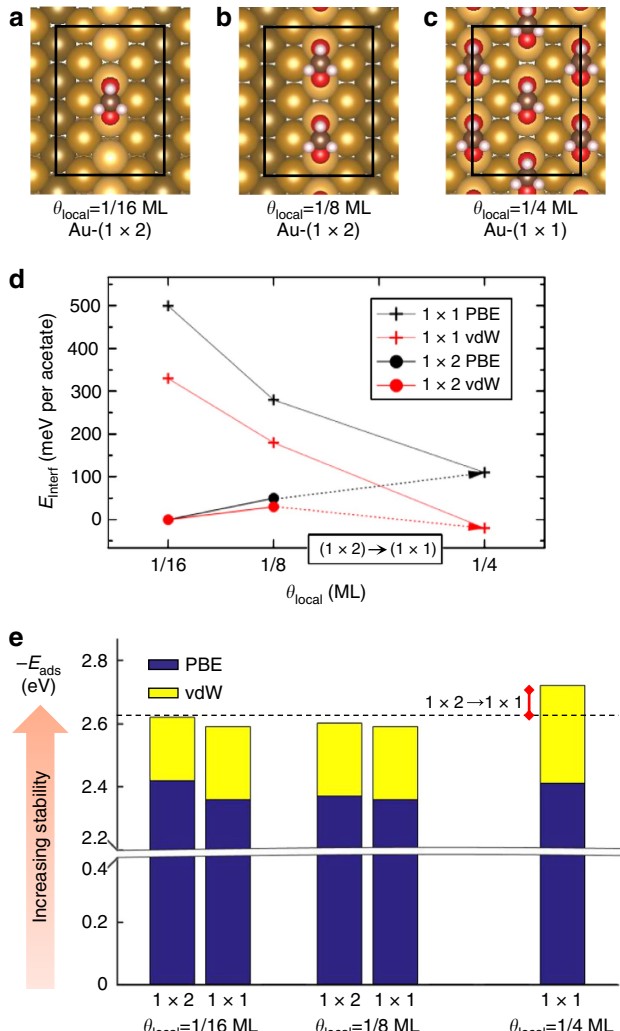

**Figure 5 | vdW interactions between adsorbed acetate molecules drive the acetate condensation and concomitant Au deconstruction.** (a–c) The most stable structures for three different local acetate coverages ($\theta_{local}$). (d) Interface energy per acetate on both Au-($1 \times 2$) and Au-($1 \times 1$) interfaces without and with vdW corrections (labelled PBE and vdW, respectively); structures with the lowest energy are the most stable. At local coverage $\theta_{local} < 1/4$ ML, acetate cannot impose the Au structural transition. The local coverage $\theta_{local} = 1/4$ ML necessitates the ($1 \times 2$)->($1 \times 1$) interface transition and is favoured with respect to lower local density structures on the ($1 \times 2$) interface only if vdW contributions are included. (e) Break down of the adsorption energy into its PBE and vdW components for all coverage-interface combinations studied. As a guide to the eye, the calculated Au-($1 \times 2$) to Au-($1 \times 1$) transformation energy is depicted by the red bar.

Note that the above considerations are dealing with the stability of large, condensed c($2 \times 2$) islands. Our calculations show that the volume energetics are favouring the condensed phase for all densities considered as the interface energy per acetate at $\theta_{local} = 1/16$ ML is already higher than interface energy per acetate in the condensed structure. The acetate-covered/clean Au interface energy comes into play in the description of the early stages of the nucleation and further investigations are underway.

Finite temperature corrections are not affecting the above conclusions. In addition to the interface energy calculated above, two contributions to the Gibbs energy of the system have to be evaluated at finite temperature (equation (4)). The configurational entropy variation $\Delta S^{conf}$ between the diluted phase

($\theta_{local} = \theta_{global} = 1/16$ ML) and the condensed phase ($\theta_{local} = 1/4$ ML) in Fig. 5a amounts to only 0.01 eV/acetate (equation (5)). Because the general adsorption geometry is calculated to be the same in the diluted and condensed phase, the variation in vibrational free energy $\Delta F^{vib}$ is expected to be negligible.

Further analysis of the adsorption of acetate reveals that the vdW contribution that drives condensation and restructuring originates from inter-adsorbate interactions occurring exclusively in the dense-packed c($2 \times 2$) structure. Here the acetate adsorption energy ($E_{ads}$) is evaluated on the various substrates and for the above-mentioned coverages (Fig. 5e). $E_{ads}$ represents the energy gain per acetate from the gas phase upon adsorption on the considered Au surface. The low-coverage vdW contribution is attributed to adsorbate-Au interaction and does not depend strongly on the Au surface reconstruction. The vdW correction represents a small portion of the acetate adsorption energy (8–11%) but this contribution per acetate is substantially enhanced (0.10 eV) in the c($2 \times 2$) structure and represents 3/2 of the contribution at lower density. This additional stabilization is attributed to inter-adsorbate interactions that occur only in the dense structure. With a density of two acetates per c($2 \times 2$) cell, which corresponds to one acetate per Au-($1 \times 2$) surface cell, the interface energy gain is comparable to the endothermicity of lifting the Au missing-row reconstruction of 0.08 eV per Au-($1 \times 2$) unit cell. Hence, the energetic cost of Au restructuring is balanced out by the stabilizing effect of the weak interactions that appear in the dense configuration. The weak interaction picture is supported by the negligible changes observed in the acetate structure throughout the systems of various densities and Au-interface structure. A detailed description of the structural changes upon adsorption is provided in the supplements (Supplementary Figs 6,7; Supplementary Tables 3–6 and Supplementary Discussion).

## Discussion

In terms of catalytic activity, beyond the poisoning effect of isolated acetate adsorption in ethanol coupling reaction (Supplementary Table 1 and Supplementary Discussion), the formation of dense acetate islands almost certainly accounts for the higher stability of acetate on gold, therefore contributing to its blocking of the whole surface sites. Additionally, the reactivity of the acetate on the surface will probably be affected by the reduction of its mobility due to self-assembly.

Although the ordering of adsorbates at surfaces and the adsorbate-induced restructuring of metal surfaces is a long known phenomenon, the underlying mechanisms have not always been determined. Interestingly, condensed carboxylate layers have been observed on other surfaces. A dense c($2 \times 2$) acetate structure was reported previously for Cu(110) (refs 24,26) and Ni(110) (refs 27,28). However, the driving force for assembly and metal atom incorporation—in a somewhat different fashion than for acetate on Au(110)—was not investigated in the mentioned studies. Weak inter-adsorbate interactions may play a role in the condensation of acetate on those surfaces as well.

Regarding adsorbate-induced restructuring, the adsorption of CO on Pt(110) is a well-known system in catalysis for which a detailed restructuring mechanism has been proposed, which is fundamentally different from the one identified here for acetate. Similarly, adsorption of CO deconstructs the missing-row on Pt(110) (refs 38,39). However, the ordered Pt-($1 \times 1$) interface with low step density can only be obtained at the saturation coverage of one CO per surface atom[40], whereas the annealing experiment presented here shows large ordered domains below the global 0.25 ML saturation coverage. In this system, as well as for high pressure CO on Au(110) (ref. 31), the identified driving

force for the surface restructuring is the creation of lower coordination atoms on which adsorbate binding is thermodynamically favoured.

In other contexts, self-assembly of molecules at surfaces can be desirable and understanding the underlying mechanisms is the first step towards its control[41,42]. While the importance of vdW inter-adsorbate interactions is well known in self-assembly of long-chain (more than six carbons) alkyl thiols on Au, here we have shown the importance of weak interactions in promoting 3D self-assembly and surface rearrangement for much smaller, strongly bound molecules. An additional remarkable feature is the dense order observed already at low global coverage.

Finally, our study is an example where weak vdW interactions need to be incorporated in the computational framework to correctly reproduce the experimentally observed phenomena. The fact that this is the case for a strongly bound chemisorbed adsorbate where vdW interactions represent only a small fraction of the adsorption energy is particularly noteworthy. In the PBE + vdW framework, intermolecular vdW interactions tip the balance towards condensation and restructuring, in agreement with the experimental observations. While the pure PBE functional may already include part of the non-local interactions in certain systems[43], it does not reproduce the trend for condensation as this transformation would be uphill in energy. In the acetate case, we compare various densities of the same adsorbed molecule and our detailed structure analysis (Supplementary Fig. 7 and Supplementary Table 6) shows very little configurational variation among the supercells considered. Therefore, we expect even higher accuracy of the computational method in comparing supercell energies.

In conclusion, we show that a combination of strong adsorbate–substrate and weak inter-adsorbate interactions leads to the formation of dense acetate islands on Au(110). The associated Au restructuring from Au-$(1 \times 2)$ to Au-$(1 \times 1)$ where the missing-rows are filled provides the necessary increase in the adsorption site density to switch on the weak acetate–acetate interactions, already at low coverage of acetate on the surface. Using the state-of-the-art DFT calculations with vdW corrections, we demonstrate the critical role of vdW interactions in driving the surface Au displacement. Hence, the present study provides a mechanism of surface restructuring, which is facilitated by the non-covalent collective effects of surface adsorbates and is related to the thermodynamics of the system. These findings may have broad implications in the structural control of surface morphology and the self-assembly of surface adsorbates and stress the counterintuitive importance of vdW corrections in DFT simulations in the case of small chemisorbed adsorbates.

## Methods

**Sample cleaning.** Two different Au(110) single crystals (Princeton Scientific) were mounted in two separate chambers dedicated to STM and temperature programmed reaction spectroscopy (TPRS). In both cases, samples were cleaned via cycles of sputtering and annealing at 800–900 K until a uniform Au(110)-$(1 \times 2)$ (Au-$(1 \times 2)$) surface was detected via STM and/or a clean low-energy electron diffraction pattern was observed.

**Oxygen adsorption and dosing of acetic acid.** For STM experiments, atomic O was adsorbed on the surface by decomposing ozone at 300 K (refs 44,45). The ozone generator (Ozone Engineering, model LG-7) was set to a concentration of 60 g N m$^{-3}$. During deposition, the chamber background pressure was in the range of $1.0$–$5.0 \times 10^{-7}$ mbar for 5–10 min. For TPRS experiments, O was deposited by thermal decomposition[46–48]. Prior to acetic acid oxidation experiments, the samples were mildly annealed to ∼ 400–450 K to equilibrate the O adatoms. While no absolute measurement of the temperature was accessible in the STM chamber, we confirm that annealing was below the onset of oxygen recombination at 450 K (ref. 49). The STM O coverage was determined from STM imaging prior to exposure to acetic acid, following the adsorption structure model discussed in ref. 22. Coverages are given with respect to the Au-$(1 \times 2)$ unit cell; 1 ML corresponds to 4 atoms per unit cell. The TPRS coverage of O on Au was calibrated

using the area under the O$_2$ recombination TPRS peaks[49]. Acetic acid (Sigma-Aldrich 99.99% purity) vapour was introduced using a directed doser through a leak valve.

**STM and TPRS experiments.** STM experiments were conducted under base pressure $< 1.0 \times 10^{-10}$ mbar using an Omicron VT-STM and mechanically cut Pt/Ir tips purchased from Veeco Instruments, Inc. Images were acquired using sample bias ranging from $+ 0.5$ to $+ 1.5$ V and tunnelling current between 0.1 and 1.0 nA with scanning speed of 100 to 500 nm s$^{-1}$; the contrast on acetate does not strongly depend on the imaging parameters. TPRS experiments were conducted following the method described in detail elsewhere, using the same experimental set-up[50].

**Computational package.** DFT calculations were performed using Vienna *ab initio* simulation package[51]. Projector augmented wave method[52] was used with a plane-wave basis set (kinetic energy cutoff 400 eV) and PBE exchange-correlation functional[53]. Dispersion interactions were approximated with the Tkatchenko–Scheffler method[54] that has been shown to yield good agreement with experiments of adsorption systems[55]. With this level of theory, the lattice constant of bulk gold was calculated as 4.11 Å (4.16 Å without dispersion correction), which is close to the experimental value (4.08 Å) (ref. 56) and previous calculation results[57].

**Modelling of the surface.** The Au(110) surface is modelled by slabs of six atomic layers whose two bottom layers are constrained to the bulk positions. More than 10 Å of vacuum layer was added above the adsorbates. $4 \times 2$ supercells with respect to the Au-$(1 \times 1)$ (see Fig. 5 and Supplementary Fig. 5) surface are used with a $7 \times 7 \times 1$ Gamma-centred $k$-point mesh. The positions of unconstrained atoms were relaxed to a force threshold of 0.01 eV Å$^{-1}$. In that scheme, the relaxed bare surface structure (Supplementary Fig. 6 and Supplementary Table 3) as well as the energetics of reconstruction are consistent with previous studies[36,58].

**STM image simulation.** STM images were simulated on the basis of the Tersoff–Hamann approximation[59]. They consist of maps of local density of states integrated over the energy window matching the experimental conditions at a constant height above the surface.

**Quantifying acetate stability.** To investigate acetate condensation, a descriptor has to be constructed to allow meaningful comparison of the energetics of systems where the local acetate density as well as the Au interface structure and total number of Au atoms is varied. The following describes the construction of the interface energy per acetate used in Fig. 5d.

First, the interface energy per supercell is defined as:

$$E_{\text{interf}, \theta_{\text{local}}, (1 \times \alpha)} = E_{\text{tot}, \theta, (1 \times \alpha)} - n_{\text{Au} - (1 \times \alpha)} \times \mu_{\text{Au, bulk}} - n_\theta \times \mu_{\text{acetate}} \quad (1)$$

with $\theta_{\text{local}} = 1/16$, 1/8 and 1/4 ML the supercell acetate coverage, $(1 \times \alpha)$ with $\alpha = 1,2$ the considered Au interface, $E_{\text{tot}}$ the total energy of the supercell considered, $n_{\text{Au}}$ the number of Au atoms in the supercell, $n_\theta$ the number of acetate molecules and $\mu_{\text{Au}}$ and $\mu_{\text{acetate}}$ the chemical potentials of bulk Au and of acetate, respectively. All supercells considered are represented in Supplementary Fig. 5a–g.

With the aim of determining the preferred phase for the same global coverage, we construct the interface energy per acetate, with $E_{\text{interf}, \theta_{\text{local}} = 1/16 \text{ ML}, (1 \times 2)}$ as our reference:

$$\Delta E_{\text{interf, condens}}(\theta_{\text{local}}, (1 \times \alpha))$$
$$= \left( m \times E_{\text{interf}, \theta_{\text{local}}, (1 \times \alpha)} + (4 - m) \times E_{\text{interf}, \theta_{\text{local}} = 0 \text{ ML}, (1 \times 2)} - E_{\text{interf}, \theta_{\text{local}} = 1/16 \text{ ML}, (1 \times 2)} \right) / 4 \quad (2)$$

With $m$ so that $m \times \theta_{\text{local}} \times 16 = 4$. This corresponds to a linear combination of the energetics of four supercells of coverage $\theta_{\text{local}}$ and $\theta_{\text{local}} = 0$ ML while maintaining a global coverage of $\theta_{\text{global}} = 1/16$ ML, our reference. The construction for each local coverage $\theta_{\text{local}} = 1/16$, 1/8 and 1/4 ML is represented schematically in Supplementary Fig. 5h.

In Fig. 5e, the adsorption energy ($E_{\text{ads}}$) of acetate is calculated as referenced to the energy of the clean Au(110) surface with respective reconstruction, such that:

$$E_{\text{ads}} = \frac{1}{N_{\text{acetate}}} [(E_{\text{tot}}) - (E_{\text{slab}}) - N_{\text{acetate}}(E_{\text{acetate}})], \quad (3)$$

where $E_{\text{tot}}$ is the total energy of the supercell, which contains the Au(110) slab and $N_{\text{acetate}}$ acetate adsorbates; $E_{\text{slab}}$ is the energy of the relaxed Au(110) slab without adsorbates, and $E_{\text{acetate}}$ is the energy of an isolated charge-neutral acetate.

**Finite temperature and pressure corrections.** At finite temperature $T$ and pressure $p$, an atomistic thermodynamics framework can be built using the Gibbs free energy:

$$G(T, p) = E^{\text{tot}} + F^{\text{vib}} - TS^{\text{conf}} + pV, \quad (4)$$

where $E^{\text{tot}}$ is the DFT-calculated total energy, $F^{\text{vib}}$ is the vibrational free energy contribution and $S^{\text{conf}}$ is the configurational entropy. $S^{\text{conf}}$ is related to the

configurations the adsorbates can adopt on the surface and its variation is calculated as follows:

$$T\Delta S^{conf} = T \times k_B \Delta[\ln(W)], \tag{5}$$

where $k_B$ is the Boltzmann constant and $W$ is the number of possible configurations.

**Data availability.** The authors declare that the data supporting the findings of this study are available within the article and its Supplementary Information files, and all relevant data are available from the authors.

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

## Acknowledgements

We thank M.M. Montemore for helpful discussions. This work was supported as part of the Integrated Mesoscale Architectures for Sustainable Catalysis (IMASC), an Energy Frontier Research Center funded by the U.S. Department of Energy (DOE), Office of Science, Basic Energy Sciences (BES), under Award number DE-SC0012573. W.C. was partially supported by the National Natural Science Foundation of China (Grant No. 11504357). The computations in this paper were run on the Odyssey cluster supported by the FAS Division of Science, Research Computing Group at Harvard University and on the Oak Ridge Leadership Computing Facility (OLCF) and the National Energy Research Scientific Computing Center (NERSC) of the U.S. Department of Energy.

## Author contributions

C.M.F., R.J.M. and E.K. guided the research. F.H. designed, conducted and analysed the microscopy experiments, B.S. the thermal desorption experiments. W.C. and B.S. designed and performed the DFT calculations, and W.C., B.S. and F.H. analysed the results. F.H. drafted the article, and all authors contributed to the final manuscript.

## Additional information

**Competing financial interests:** The authors declare no competing financial interests.

