## [Peer Review File · Nature Communications]

Reviewers' comments:

Reviewer #1 (Remarks to the Author):

The authors present a combined STM and DFT study of the lifting of the Au(110) surface reconstruction by adsorbed acetate. While this is well-performed solid work, it is not apparent that the results represent a major breakthrough in this mature field and thus fulfill the requirements of novelty and importance. Adsorbate-induced restructuring of metal surfaces has been extensively studied in the past. Arguably the most important case is the transition between the (1x2) missing row reconstructed (110) surface of noble and transition metals in the presence of a large number of adsorbate species, including acetate. Many of the observations and conclusions presented in the manuscript have already been reported in previous studies.

The STM observations strongly resemble previous results where similar restructuring mechanisms on terraces and along steps were found. Specifically, the "bimodal roughening" in form of highly anisotropic islands has been already observed in the 1990ies. Also the effect of annealing is not surprising, since the restructuring kinetics strongly depends on the surface mobility and the formation of large defect-free islands will always be energetically favorable in the (1x1) phase.

The main new result according to the authors is the influence of the van-der-Waals between the adsorbates on the restructuring process. However, it is well known that lifting of surface reconstructions occurs in many cases only above a threshold coverage. That the surface restructuring involves collective effects rather than the interaction of an isolated adsorbate with the surface is not a new idea. What actually is novel to my knowledge is that the authors quantify this effect for the present system by DFT calculations and show that vdW interactions have to be taken into account to explain this behavior. This may be the most interesting point of the paper, but the differences in energy are rather small and it is not clear whether these results depend on the specific functional employed in this work.

If the structure-decisive effect of the lateral adsorbate interactions can be unambiguously proven, the paper may be of sufficient interest to the surface science and catalysis community to warrant publication. It could perhaps be strengthened by focusing on this topic and reducing the detailed description of the restructuring mechanisms. The later are not that novel and are not essential for the discussion of the driving force of the restructuring.

As a more minor point, the authors state on p. 9 that the final distribution of acetate on the surface is much different from the distribution of atomic O. They either should give experimental evidence for this or remove this statement.

Reviewer #2 (Remarks to the Author):

A. Summary of the key results

A: this is done in the abstract, but a Conclusions section summarizing the key results is missing

B. Originality and interest: if not novel, please give references

B: this work is novel and of interest to the catalysis community, showing the importance of van der Waals interactions in the formation of SAMs of acetate on Au(110)

C. Data & methodology: validity of approach, quality of data, quality of presentation

C: the paper combines STM with DFT, a valuable approach to the system studied. STM only could never explain the observations, while DFT in this case is able to proof the importance of vdW interactions in this system

D. Appropriate use of statistics and treatment of uncertainties

D: not applicable

E. Conclusions: robustness, validity, reliability

E: No conclusions presented, these should be added before publication

F. Suggested improvements: experiments, data for possible revision

F:

1) page 5, lines 3,4: the fact that a local coverage of 0.25 ML is observed, while the global coverage is only 0.05 ML implies that you also should observe areas of bare Au(110). Is this indeed the case? And do these areas amount to the expected 80% of the surface? Please show this with STM images.

2) page 6, line 10: please show image of bare Au(110) to show roughening of the surface

3) page 7, line 8: is this terminal atom Au? Please clarify

4) starting from page 9, line 14, the authors are continuously referring to Fig 3, whereas this should be Fig. 4. Please change this everywhere in the text where this is applicable

5) page 11, line 21: absorption should be adsorption

6) Discussion consists of three statements that seem not to be related. Please rewrite such that it has more relation and also better connection to the results section

7) SI: order is not logical, the first figure is not discussed until late in the main manuscript

8) SI page 6: conversion of 0.6 eV to 60 kJ/mol is quite a rough estimate, why not use 58 kJ/mol

9) SI page 7: S5a,b are not experimental STM images

G. References: appropriate credit to previous work?

G: ok

H. Clarity and context: lucidity of abstract/summary, appropriateness of abstract, introduction and conclusions

H: abstract is fine, introduction is fine. Conclusions section is missing

Reviewer #3 (Remarks to the Author):

Adsorbate induced restructuring of a metal surface has been the subject of many experimental and theoretical studies for the past thirty years, as the authors are aware. Acetate induced restructuring of a number of transition metal surfaces has also been broadly studied - much owed to the pioneering work done by the senior author (and his co-workers) of the present manuscript. The role of van der Waals interaction as being essential for proper description of both acetate-acetate interactions and acetate-metal-surface interaction has also been the subject of a number of publications, some already referred in the present manuscript. To the authors' credit, relevant publications have been cited in the manuscript, so I do not feel the need to provide more.

The question then is what is new from this work?

I see two main contributions:

Very elaborate and quantitative experimental observations using STM that show that acetate molecules when adsorbed on Au(110) condense into structures with high local concentration even when the global concentration is low. The evidence for this comes from pronounced surface roughening which the authors ascribe to a process in which Au atoms from the clean surface participate in the self-assembly of the acetate molecules. The STM images are clear and suggestive of strong adsorbate induced restructuring of the Au surface by acetate molecules.

The second main result comes from the accompanying DFT calculations which validate the thermodynamic stability of the proposed structures and are able to do so only when they include van der Waals interactions. This is an interesting result and even if one argues about the validity of the different functionals used in DFT and DFT with van der Waals correction, the results are a good example of the importance of inclusion of van der Waals interactions in such calculations.

In summary, this is a nice piece of work on a subject that has been the subject of extended experimental and theoretical investigations for several decades. The availability of more refined experimental and theoretical tools has allowed the authors to shine some more light on topics that has puzzled some of them for a long time.

Response to Reviewers' comments:

Response to Reviewer #1:

Reviewer's comment: The authors present a combined STM and DTF study of the lifting of the Au(110) surface reconstruction by adsorbed acetate. While this is well-performed solid work, it is not apparent that the results represent a major breakthrough in this mature field and thus fulfill the requirements of novelty and importance. Adsorbate-induced restructuring of metal surfaces has been extensively studied in the past. Arguably the most important case is the transition between the (1x2) missing row reconstructed (110) surface of noble and transition metals in the presence of a large number of adsorbate species, including acetate.

Many of the observations and conclusions presented in the manuscript have already been reported in previous studies. The STM observations strongly resemble previous results where similar restructuring mechanisms on terraces and along steps were found. Specifically, the "bimodal roughening" in form of highly anisotropic islands has been already observed in the 1990ies.

Response:

The work presented in our paper is distinct from previous reports because we demonstrate that the restructuring mechanism relies on noncovalent interactions (vdW) between adsorbates to drive the reconstruction. This is supported both by the detailed experimental analysis and the DFT calculations.

We have modified the text so as to include a more detailed discussion that more clearly delineates our work from previous studies of surface reconstruction. We have added the following:

On P7: "The step retraction and row-pairing mechanisms are similar to the deconstruction mechanism of the metastable missing-row structure on Rh(110),²⁹ including the important role of defects such as under-coordinated metal atoms at kinks within step edges in the structural transition.³⁰ It is however important to note that in the cited study, the surface is adsorbate-free and the thermodynamics of the metal surface itself is the driving force for the reconstruction."

The CO/Pt and Acetate/Cu(110) examples are discussed in the revised paragraphs 2 and 3 of the discussion.

"Although the ordering of adsorbates at surfaces and the adsorbate-induced restructuring of metal surfaces is a long known phenomenon, the underlying mechanisms have not always been determined. Interestingly, condensed carboxylate layers have been observed on other surfaces. A dense c(2x2) acetate structure was reported previously for Cu(110),^{24,26} and Ni(110).^{27,28} However, the driving force for assembly and metal atom incorporation –in a somewhat different fashion than for acetate on Au(110)- was not investigated in the mentioned studies. Weak inter-adsorbate interactions may play a role in the condensation of acetate on those surfaces as well.

Regarding adsorbate-induced restructuring, the adsorption of CO on Pt(110) is a well-known system in catalysis for which a detailed restructuring mechanism has been proposed which is fundamentally different from the one identified here for acetate. Similarly, adsorption of CO deconstructs the missing row on Pt(110).^{38,39} However, the ordered Pt-(1x1) interface with low step density can only be obtained at the saturation coverage of one CO per surface atom⁴⁰ whereas the annealing experiment presented here shows large ordered domains below the global 0.25 ML saturation coverage. In this system, as well as for high pressure CO on Au(110),³¹ the identified driving force for the surface restructuring is the creation of lower coordination atoms on which adsorbate binding is thermodynamically favored."

The added conclusion emphasizes the novelty of our conclusions:

"In conclusion, we show that a combination of strong adsorbate-substrate and weak inter-adsorbate interactions leads to the formation of dense acetate islands on

Au(110). The associated Au restructuring from Au-(1x2) to Au-(1x1) where the missing rows are filled provides the necessary increase in the adsorption site density to switch on the weak acetate-acetate interactions, already at low coverage of acetate on the surface. Using the state-of-the-art DFT calculations with vdW corrections, we demonstrate the critical role of vdW interactions in driving the surface Au displacement. Hence, the present study provides a novel mechanism of surface restructuring, which is facilitated by the non-covalent collective effects of surface adsorbates and is related to the thermodynamics of the system. These findings may have broad implications in the structural control of surface morphology and the self-assembly of surface adsorbates and stress the – counterintuitive- importance of vdW corrections in DFT simulations in the case of small chemisorbed adsorbates.”

Reviewer’s comment:

Also the effect of annealing is not surprising, since the restructuring kinetics strongly depends on the surface mobility and the formation of large defect-free islands will always be energetically favorable in the (1x1) phase.

Response:

The annealing experiment proves that the trend is to form large dense islands and reduced density of steps, eliminating the possible interpretation of a stronger molecule-substrate interaction with undercoordinated metal atoms, which is the underlying mechanism proposed for surface restructuring for CO/Pt(110) (Thostrup et al., PRL 87, 126102 (2001)). Notably, in the above study, it is shown that the surface still exhibits the bimodal surface even after annealing at 400K. Hence, large defect-free islands are not always favored.

P9, the following text was added at the end of the paragraph presenting the annealing experiment:

“Closer to the thermodynamic equilibrium, large flat acetate domains are formed, even for a partially-covered surface. Hence, the annealing experiment definitely eliminates the possibility of stronger molecule-substrate interaction with undercoordinated metal atoms as the driving force for reconstruction.”

The case of CO/Pt and how it differs from our study is discussed in detail in the discussion, as mentioned above.

Reviewer’s comment:

The main new result according to the authors is the influence of the van-der-Waals between the adsorbates on the restructuring process. However, it is well known that lifting of surface reconstructions occurs in many cases only above a threshold coverage. That the surface restructuring involves collective effects rather than the interaction of an isolated adsorbate with the surface is not a new idea.

Response:

Although the reviewer did not elaborate on what is meant by “collective effects” we think it pertains to a critical coverage, a notion indeed well-known in nucleation theory, but distinct from our results. In our work, we provide insight into the underlying reason for the restructuring—weak inter-adsorbate interactions. This is achieved by compensating the

restructuring cost through stabilizing interadsorbate interaction. In the conclusion, we added the following:

“Hence, the present study provides a novel mechanism of surface restructuring, which is facilitated by the non-covalent collective effects of surface adsorbates and is related to the thermodynamics of the system.”

Reviewer’s comment:

What actually is novel to my knowledge is that the authors quantify this effect for the present system by DFT calculations and show that vdW interactions have to be taken into account to explain this behavior. This may be the most interesting point of the paper, but the differences in energy are rather small and it is not clear whether these results depend on the specific functional employed in this work.

Response:

- Regarding the magnitude of the differences in energy:

First, the energies presented in Fig. 5 were normalized per acetate. The actual energy difference for the “isolated acetate” to c(2x2) supercell (containing four acetates) is +0.45 eV with PBE only and -0.08 eV with the correction PBE+vdW, highlighting the collective stabilization effect. Those values are well above error-bars for such calculations and therefore we believe they are reliable.

On Page 11, we have added the sentence: “In terms of supercell energy i.e. 4 acetate molecules, this corresponds to +0.45 eV uphill with PBE only and -0.08 eV downhill with the corrected PBE+vdW, revealing their importance in describing the thermodynamics of the system.”

- Regarding the functional reliability:

Second, indeed one has to be careful when comparing energies of different systems using DFT methods. However, in our case, the system is very similar in all supercells. Our detailed structure analysis (Supplements, Fig. S7 and Table S6) shows very little configurational variations among the various densities of acetate and Au interface. Therefore, we expect even higher accuracy of the method in comparing supercell energies than for the adsorption of different adsorbates, which show good agreement with experiments (see below).

We have added the above remark in the discussion part of the manuscript.

Based on our own experience, the DFT-TS method we used here accurately describes similar surface-adsorption systems. [REDACTED]

According to a recent review article, vdW correction functionals, semiclassical C6 based and nonlocal density based, “yield equally high accuracy for various noncovalent interaction motifs.”*

- *Dispersion-Corrected Mean-Field Electronic Structure Methods. Grimme et al. Chem. Rev. 2016, 116, 5105–5154.*

Reviewer's comment:

If the structure-decisive effect of the lateral adsorbate interactions can be unambiguously proven, the paper may be of sufficient interest to the surface science and catalysis community to warrant publication. It could perhaps be strengthened by focusing on this topic and reducing the detailed description of the restructuring mechanisms. The later are not that novel and are not essential for the discussion of the driving force of the restructuring.

Response:

The description of the restructuring mechanisms has been reduced according to the referee's suggestion, however we would like to emphasize that the detailed analysis of the experimental data is essential to provide a proof of (1) the interface structure by following the movement of Au atoms in order to form the Au(110) -1x1 interface and (2) support the mechanism unveiled by state-of-the-art calculations because it shows dense structures of 0.25 ML local coverage already at low global coverage.

Reviewer's comment:

As a more minor point, the authors state on p. 9 that the final distribution of acetate on the surface is much different from the distribution of atomic O. They either should give experimental evidence for this or remove this statement.

Response:

We have characterized the adsorption of atomic O in detail previously in a published paper. A reference to the published study has been added. (Direct visualization of quasi-ordered oxygen chain structures on Au(110)-(1x2). Surf. Sci. 650, 5 (2016)). Additionally, an inset in Fig.2a with the initial O-covered surface has been added.

Response to Reviewer #2

A. Summary of the key results

A: this is done in the abstract, but a Conclusions section summarizing the key results is missing

A separate conclusion has now been added.

B. Originality and interest: if not novel, please give references

B: this work is novel and of interest to the catalysis community, showing the importance of van der Waals interactions in the formation of SAMs of acetate on Au(110)

No response required.

C. Data & methodology: validity of approach, quality of data, quality of presentation

C: the paper combines STM with DFT, a valuable approach to the system studied. STM

only could never explain the observations, while DFT in this case is able to proof the importance of vdW interactions in this system

No response required.

D. Appropriate use of statistics and treatment of uncertainties

D: not applicable

No response required.

E. Conclusions: robustness, validity, reliability

E: No conclusions presented, these should be added before publication

A separate conclusion has been added.

F. Suggested improvements: experiments, data for possible revision

F: 1) page 5, lines 3,4: the fact that a local coverage of 0.25 ML is observed, while the global coverage is only 0.05 ML implies that you also should observe areas of bare Au(110). Is this indeed the case? And do these areas amount to the expected 80% of the surface? Please show this with STM images.

Response:

The coverage is actually calculated based on the area covered by acetate and the local coverage within the acetate island, which is obtained from the high-resolution images (Fig. 2d,e). We have now modified the paper on p. 5 to make this point clearly by adding the following:

“The local coverage that corresponds to this molecular arrangement is 0.25 ML. It is noteworthy that this dense ordering is observed even at global coverage as low as 0.05 ML, which is derived from the ratio of clean Au area (row contrast) and acetate-covered areas (bright domains) (Fig. 2a). The details of the c(2x2) are apparent in high resolution images (Fig. 2d-e).”

2) page 6, line 10: please show image of bare Au(110) to show roughening of the surface

Response:

We have added the following to address this point in a way that does not add additional figures:

P6: The terraces and step edges roughen upon acetate adsorption, as can be seen from the STM image series at increasing coverage in Fig. 2a-c. This indicates the mobilization of Au atoms in the acetate condensation process.”

Additionally, we have highlighted the word “under” in the following sentence in order to clearly state that the roughening occurs from the acetate/Au interface.

P6: “The bimodal roughening of the surface can be interpreted as the transition from “missing row” to ideal Au interface structure under the acetate layer”

3) page 7, line 8: is this terminal atom Au? Please clarify

Response:

We have modified the sentence: “the terminal Au atom migrates from the step kink onto the upper terrace”.

4) starting from page 9, line 14, the authors are continuously referring to Fig 3, whereas this should be Fig. 4. Please change this everywhere in the text where this is applicable

Response:

On P10, 17, 18, we have changed Figs. 4 to Figs. 5.

5) page 11, line 21: absorption should be adsorption

We have made the suggested change.

6) Discussion consists of three statements that seem not to be related. Please rewrite such that it has more relation and also better connection to the results section

We rewrote the discussion. It is now composed of a part dealing with the results in the context of catalysis, followed by a discussion of the restructuring mechanism observed and how it differs from some well documented systems. Finally, we discuss the implications of our findings in the context of DFT simulations of molecules on surfaces.

7) SI: order is not logical, the first figure is not discussed until late in the main manuscript

Response:

The figure is now right after Fig. S5, figures have been renumbered both in main text and supplements.

8) SI page 6: conversion of 0.6 eV to 60 kJ/mol is quite a rough estimate, why not use 58 kJ/mol

Response:

We have made the suggested change.

9) SI page 7: S5a,b are not experimental STM images

We have changed Fig. S5a,b to Fig. S5c,d.

G. References: appropriate credit to previous work?

G: ok

No response required.

H. Clarity and context: lucidity of abstract/summary, appropriateness of abstract, introduction and conclusions

H: abstract is fine, introduction is fine. **Conclusions section is missing**

Response:

A separate conclusion has been added.

Response to Reviewer #3:

Reviewer's comment:

Adsorbate induced restructuring of a metal surface has been the subject of many experimental and theoretical studies for the past thirty years, as the authors are aware. Acetate induced restructuring of a number of transition metal surfaces has also been broadly studied - much owed to the pioneering work done by the senior author (and his co-workers) of the present manuscript. The role of van der Waals interaction as being essential for proper description of both acetate-acetate interactions and acetate-metal-surface interaction has also been the subject of a number of publications, some already referred in the present manuscript. To the authors' credit, relevant publications have been cited in the manuscript, so I do not feel the need to provide more.

The question then is what is new from this work? I see two main contributions: Very elaborate and quantitative experimental observations using STM that show that acetate molecules when adsorbed on Au(110) condense into structures with high local concentration even when the global concentration is low. The evidence for this comes from pronounced surface roughening which the authors ascribe to a process in which Au atoms from the clean surface participate in the self-assembly of the acetate molecules. The STM images are clear and suggestive of strong adsorbate induced restructuring of the Au surface by acetate molecules.

The second main result comes from the accompanying DFT calculations which validate the thermodynamic stability of the proposed structures and are able to do so only when they include van der Waals interactions. This is an interesting result and even if one argues about the validity of the different functionals used in DFT and DFT with van der Waals correction, the results are a good example of the importance of inclusion of van der Waals interactions in such calculations.

In summary, this is a nice piece of work on a subject that has been the subject of extended experimental and theoretical investigations for several decades. The availability of more refined experimental and theoretical tools has allowed the authors to shine some more light on topics that has puzzled some of them for a long time.

Response:

We thank the reviewer for the thoughtful analysis and positive comments. No response required.

We have also addressed all points related to the editorial policies:

- *Completed authors' postal addresses*
- *Reordered sections*
- *Italicized scalar variables*
- *Reorder paragraphs in the Introduction (summary of results at the end)*

- *Shorten some subtitles (60 characters limit)*
- *Fixed supplementary reference format*
- *Modified reference formatting (end page)*
- *Added authors contributions*
- *Added conflict of interest statement*
- *Removed numbers on scale bars in Figures*
- *All the supplementary items are referenced in the main text*

We have now made significant modifications to the paper to address the specific comments by the reviewers and have strengthened the paper by clarification and more concise writing. The reviewers' comments overall were helpful in making the paper more impactful. Thank you for your consideration.

Sincerely,

Cynthia M. Friend
T.W. Richards Professor of Chemistry
Professor of Materials Science
Director, Rowland Institute at Harvard
Director, IMASC Energy Frontier Research Center

REVIEWERS' COMMENTS:

Reviewer #1 (Remarks to the Author):

The changes made by the authors definitely have improved the manuscript. In particular, the reduction of the part dealing with the surface restructuring and the stronger focus on the origin of this restructuring help to bring out the central result more clearly. In the present form the paper should be of interest to a wider community in the area of catalysis and surface science.

Minor comment:

p.9, l. 4: This probably refers to Fig. 4a not Fig. 3a.

Reviewer #2 (Remarks to the Author):

The authors have changed their manuscript significantly incorporating the suggestions of all three referees. I am satisfied with the improvements made and conclude that the manuscript is now suitable for publication in Nature Communications in its present form.

The reviewers only requested minor changes that have been addressed. The reviewers' comments are reported below and our comments are in italics.

REVIEWERS' COMMENTS:

Reviewer #1 (Remarks to the Author):

The changes made by the authors definitely have improved the manuscript. In particular, the reduction of the part dealing with the surface restructuring and the stronger focus on the origin of this restructuring help to bring out the central result more clearly. In the present form the paper should be of interest to a wider community in the area of catalysis and surface science. Minor comment:

p.9, l. 4: This probably refers to Fig. 4a not Fig. 3a.

Correct, we have made the corresponding change.

Reviewer #2 (Remarks to the Author):

The authors have changed their manuscript significantly incorporating the suggestions of all three referees. I am satisfied with the improvements made and conclude that the manuscript is now suitable for publication in Nature Communications in its present form.

No change requested